# Complementary resource preferences spontaneously emerge in diauxic microbial communities

Zihan Wang[1,2,7], Akshit Goyal[3,7✉], Veronika Dubinkina [2,4], Ashish B. George[2,5], Tong Wang [1,2], Yulia Fridman[6] & Sergei Maslov [1,2,4✉]

Many microbes grow diauxically, utilizing the available resources one at a time rather than simultaneously. The properties of communities of microbes growing diauxically remain poorly understood, largely due to a lack of theory and models of such communities. Here, we develop and study a minimal model of diauxic microbial communities assembling in a serially diluted culture. We find that unlike co-utilizing communities, diauxic community assembly repeatably and spontaneously leads to communities with complementary resource preferences, namely communities where species prefer different resources as their top choice. Simulations and theory explain that the emergence of complementarity is driven by the disproportionate contribution of the top choice resource to the growth of a diauxic species. Additionally, we develop a geometric approach for analyzing serially diluted communities, with or without diauxie, which intuitively explains several additional emergent community properties, such as the apparent lack of species which grow fastest on a resource other than their most preferred resource. Overall, our work provides testable predictions for the assembly of natural as well as synthetic communities of diauxically shifting microbes.

[1] Department of Physics, University of Illinois at Urbana-Champaign, Urbana, IL 61801, USA. [2] Carl R. Woese Institute for Genomic Biology, University of Illinois at Urbana-Champaign, Urbana, IL 61801, USA. [3] Physics of Living Systems, Department of Physics, Massachusetts Institute of Technology, Cambridge, MA 02139, USA. [4] Department of Bioengineering, University of Illinois at Urbana-Champaign, Urbana, IL 61801, USA. [5] Department of Plant Biology, University of Illinois at Urbana-Champaign, Urbana, IL 61801, USA. [6] National Research Center "Kurchatov Institute", Moscow 123182, Russia. [7] These authors contributed equally: Zihan Wang, Akshit Goyal. ✉email: akshitg@mit.edu; maslov@illinois.edu

In many natural microbial communities, species coexist with each other despite significant overlaps in their metabolic capabilities[1–5]. Understanding how microbes can coexist despite such overlaps is a big challenge in microbial ecology[6–9]. Many theories and models have been proposed towards this end, most notable being the consumer-resource models proposed by MacArthur and since then, studied by many others[10–13]. Such consumer-resource models have been used to explain several experimentally observed properties of microbial communities, such as a high species diversity supported via the exchange of metabolic byproducts (cross-feeding), as well as the division of labor in multispecies communities[14–17]. However, these models often include a simplifying assumption—microbes that grow on multiple resources always utilize them simultaneously—which many microbial species violate.

Indeed, classic work on *E. coli* and *B. subtilis* by Monod established that in environments where multiple resources are present, microbes utilize them one at a time via a phenomenon called diauxie (growth in two phases)[18]. A diauxic microbial species has a particular resource preference order, ranging from a most preferred (top choice) resource to a least preferred resource. Microbes exhibit diauxic shifts for a multitude of resources, including polysaccharides in the gut and amino acids in the soil[19,20]. Importantly, not all microbes have the same diauxic preferences. Even for sugars such as glucose, which are expected to be universally prized, there are microbial species that do not prefer them the most (called reverse diauxie)[21–24]. Further work on microbes from a variety of environments, such as the human gut and soils, has confirmed that sequential utilization, or diauxie, is prevalent in microbes, being the norm rather than the exception[25–27]. However, the assembly of diauxic communities, communities of microbes that grow via diauxie, remains unknown and understudied.

Here, we propose and study a minimal model of microbial communities undergoing diauxic growth. Using the model, we find a property that dictates community assembly at balanced resource supply, that is: coexisting species almost always have complementary preferences for their top choice resources. In other words, the top choice resource of every species in a diauxic community is unique, different from every other species in the community. The spontaneous emergence of complementarity occurs only for the top choice resource. For all other choices, complementarity reduces over community assembly, as expected by chance. Using computer simulations and geometric theory, we show that the top choice resource overwhelmingly drives the growth of diauxic microbes and exerts a disproportionately strong influence on overall community assembly. The importance of the top choice resource for growth also explains another empirical observation from natural communities—the apparent lack of anomalous species, which grow fastest on a resource other than their top choice[19,28,29]. Using experimental measurements of growth parameters from 13 diauxic *E. coli* strains, we predict a variety of pairwise competitive outcomes by simulating our model.

Taken together, these results highlight a strong selection pressure that acts on microbial regulatory networks to determine their most preferred resource. Importantly, this form of selection manifests through multispecies community assembly and is thus not expected in isolated microbes.

## Results

### A model of diauxic community assembly.
Community models studying diauxie should mimic serial dilution cultures instead of chemostats, in order to make their predictions both experimentally and ecologically relevant. Experimentally, microbial community assembly assays frequently utilize serial dilution cultures. Ecologically, diauxic growth is best suited to a "feast and famine" lifestyle, which a serial dilution culture mimics[30–32]. Therefore, throughout this manuscript, we model the assembly of a microbial community undergoing a sequence of growth-dilution cycles (see Fig. 1a). Community assembly occurs gradually through the addition of microbial species from a diverse species pool one at a time. Each species in the pool consumes resources *diauxically*, i.e., one at a time according to its resource preference.

We begin by illustrating the growth of a single species (labeled $\alpha$) grown in an environment with four resources (Fig. 1a–c). The species first grows on its most preferred resource ($R_1$) with a growth rate $g_{\alpha 1}$ until time $T_1$, when this resource gets exhausted. After a lag period $\tau$, the species switches to growing on its next preferred resource ($R_3$) with growth rate $g_{\alpha 3}$ until time $T_3$, when this resource also gets exhausted. This process of diauxic growth by sequential utilization of resources continues until either all resources are depleted, or the cycle ends at time $T$. At this point, a fraction $1/D$ of the medium containing the species is transferred to a fresh medium replete with resources. This corresponds to the dilution of species abundances by a factor $D$, mimicking serial dilution experiments in the laboratory.

After several transfers, species dynamics converge to a steady state, where each species starts a cycle with the same initial abundance as the previous cycle. At this point, we add a small population of a new invader species, chosen randomly from the species pool, to the steady-state community (Fig. 1d, e). (Hence, we assume that species invasions are rare enough such that communities always reach a steady state before the next invasion.) The invader may differ from the resident species in both resource preference order and growth rates on each resource (Fig. 1a). Once introduced, the invader may grow and establish itself in the community in a new steady state (Fig. 1d, e), or it may fail, returning the community to its previous steady state.

The growth rates and preference orders completely characterize a species, while the set of resource depletion times ($T_1$, $T_2$, etc.) characterize the current state of the abiotic environment. As we will later show, these resource depletion times are important observables in a community, since they determine the success or failure of an invader.

A realistic example of a community captured by our model is the human gut microbiome, specifically the assembly of primary consumers (e.g., *Bacteroides* species) on the polysaccharides (e.g., starch, cellulose, and mucin) that they consume. Here, there is a significant overlap between the metabolic capabilities of the microbes, but they nevertheless coexist. These species often consume polysaccharides diauxically, and engage in resource competition. Moreover, several of these species have different resource preferences, which others have hypothesized help them coexist[26,33].

Throughout this paper, we neglect diauxic lag times ($\tau = 0$) for simplicity. We will later show that adding lag times only quantitatively strengthens our main results (see "Discussion" and Fig. 5). We also assume that the supplied resource concentrations are sufficiently large, enabling species to always grow exponentially at their resource-specific growth rates. Further, we assume a balanced supply of resources, i.e., that resources are supplied in equal concentrations (see "Discussion" and Supplementary Text for results in an unbalanced resource supply).

We simulated the assembly of 1000 communities, each being colonized from a pool of ~10,000 species (see "Methods"). Species could utilize all 4 supplied resources diauxically. Each species had a random resource preference order and different growth rates on each resource, which were picked randomly from a rectified normal distribution (with mean 0.25 and standard deviation

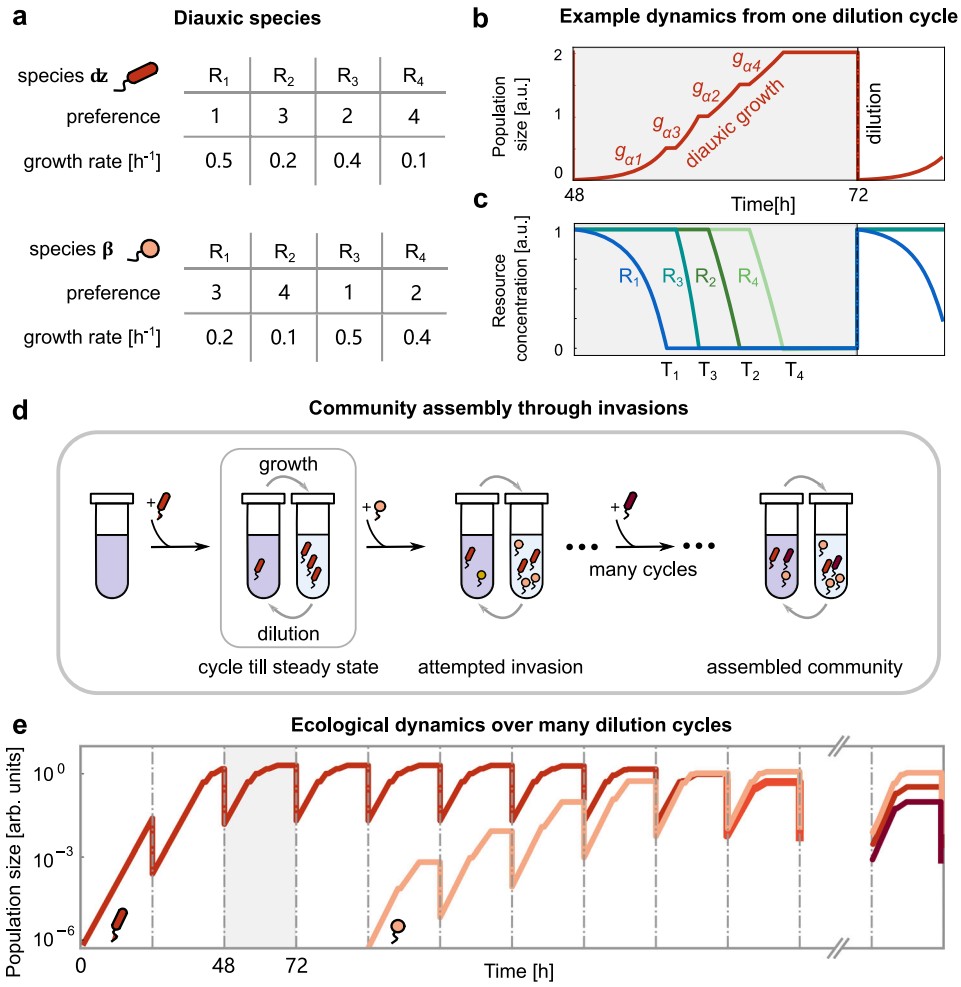

**Fig. 1 Model of community assembly with diauxie and serial dilution. a** Tables of growth rates and resource preferences of two species $\alpha$ (red) and $\beta$ (yellow), each capable of consuming all four available resources, $R_1$ to $R_4$. The resource preference sets the sequence in which a microbial species utilizes resources, and the corresponding rates $g_{Xi}$ indicate the growth rate while consuming each resource (see "Methods"). **b** Diauxic growth curve of species $\alpha$ during one serial dilution cycle, which has 4 phases of growth on each individual resource, with rates $g_{\alpha1}$, $g_{\alpha3}$, $g_{\alpha2}$, and $g_{\alpha4}$, respectively (with a brief lag period between two phases). At the end of each dilution cycle, we dilute the population by a factor $D = 100$, and supply fresh resources (see "Methods"). **c** Resource depletion curves corresponding to (**b**), where each resource is represented by a different color. $R_1$ is exhausted at time $T_1$; then species $\alpha$ consumes $R_3$ which runs out at $T_3$, which is followed by exhaustion of $R_2$ at $T_2$, and so on. **d** Schematic of serial dilution experiment. During community assembly, new species are added one by one from a species pool. After each successful invasion, the system undergoes several growth-dilution cycles until it reaches a steady state. **e** Population dynamics corresponding to the assembly process in (**d**). Panels (**b**) and (**c**) correspond to a small section of this process (highlighted in gray), where the community dynamics consist only of species $\alpha$ (red) reaching a steady state.

0.05). We assumed that the growth rate distributions for each of the 4 resources were the same, such that no resource was consistently better than the other. This is a simplifying assumption, but it nevertheless captures a variety of experimental observations showing remarkable growth rate variability of different microbial species on the same carbon sources[34–36]. Community assembly proceeded via introduction of species one at a time, in a random order, until each species had attempted to invade exactly once.

**Emergent properties of diauxic community assembly.** To study the emergent properties of communities of diauxic species, we followed the assembly process from a species pool via invasion of species one at a time. We used the number of invasion attempts to track time; communities matured over successive invasions. We found that the assembly process became slower over time—successful invasions became rarer as the community matured (Fig. 2a inset). Throughout the assembly process, we recorded

four key properties of communities: total resource depletion time, species diversity, complementarity of the community, and prevalence of anomalous species (defined below).

*Resource depletion time.* In each community, resources disappear at specific times and in a well-defined order (Fig. 1c). The total resource depletion time measures how quickly the community consumes all supplied resources. In this way, the total resource depletion time characterizes the overall speed at which a community consumes resources. The total resource depletion time decreased as communities assembled (Fig. 2a, solid line). The rate and degree of this decrease depend on the mean and variance of the growth rate distribution and the number of invasion attempts. In addition, the variability in depletion times between communities reduced over community assembly (Fig. 2a, gray lines; coefficient of variation reduces by 47%, see Fig. S1). Thus the assembly process selects for communities that collectively consume resources quickly.

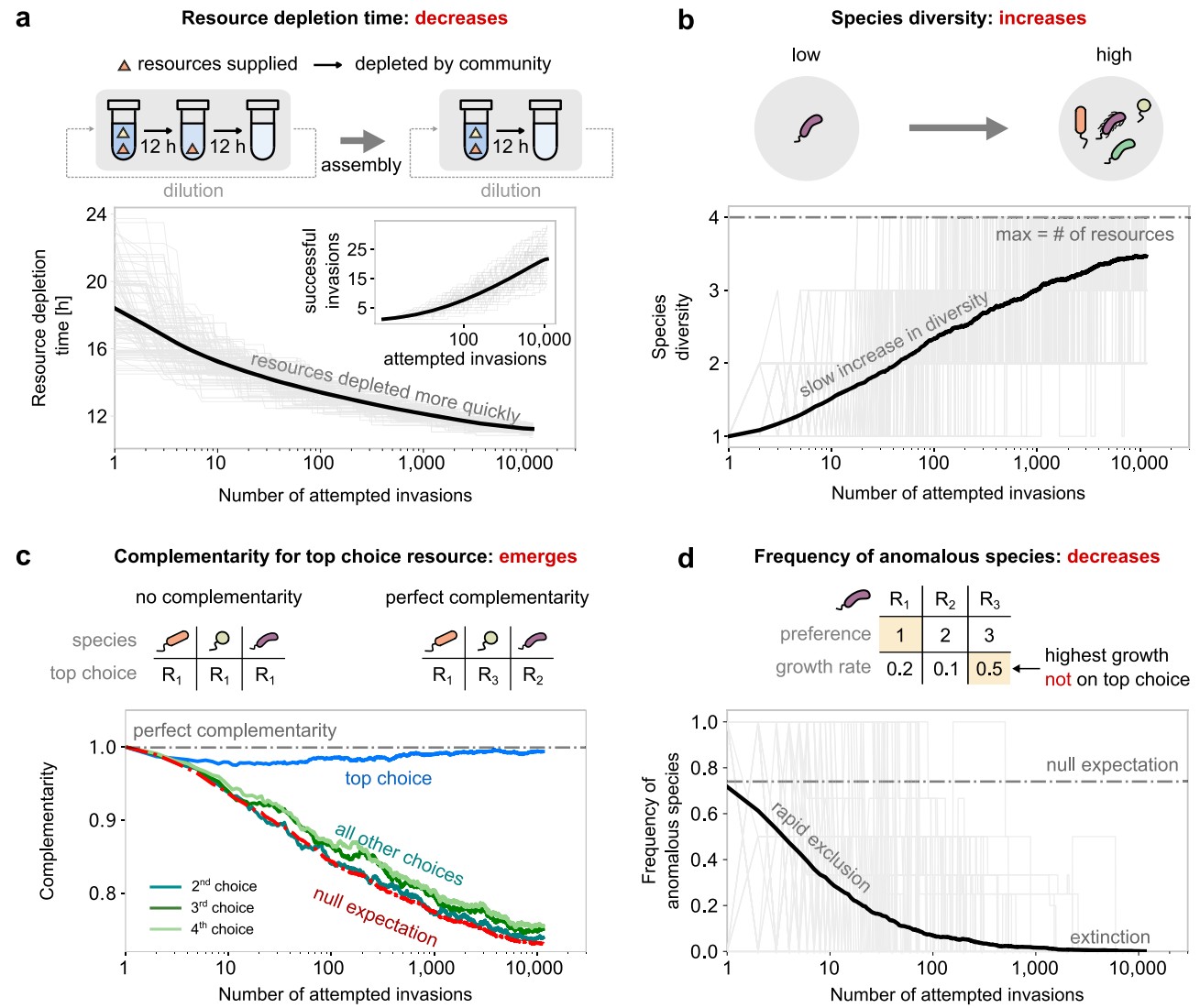

**Fig. 2 Emergent properties of diauxic microbial communities.** In all plots, solid bold lines represent the average over 958 individual community assembly simulations, while gray lines correspond to 100 randomly chosen community assembly simulations. **a** Total resource depletion time during community assembly (the time taken by the community to deplete all available resources). (Inset) Number of successful invasions during community assembly. **b** Total species diversity during community assembly (number of surviving species at steady state). **c** Resource utilization complementarity during community assembly. For each time point, the *n*th choice complementarity was calculated as a number of unique resources among the *n*-th preferred choices of all species in the community, divided by the number of unique resources in the environment. For a certain community, the null expectation (complementarity without selection) was defined by the complementarity of a random set of species from the pool that has the same diversity of that community. Colored lines show the average trend of complementarity on each preferred resource choice: top (light blue), second (cyan), third (deep green), and fourth (light green). The red dash-dotted line shows the average trend of null expectation. The gray dash-dotted line at the top corresponds to the perfect complementarity, which is 1. **d** Frequency of species with anomalous resource preferences during community assembly. The gray dash-dotted line is the expectation of fraction of anomalous species (75%) in the pools.

*Species diversity.* The species diversity was quantified as the number of species coexisting in the steady-state community. In the model, like in other consumer-resource models, the number of coexisting species at steady state is limited by the number of resources, 4 (Fig. 2b, dashed line)[10,37]. This is a natural consequence of competition for resources in our model (see Supplementary Text, section F for a derivation). Notably, species with the same resource preferences can coexist in the model, as long as the number of species is less than the number of resources (e.g., pairs of *E. coli* strains can coexist in media with glucose and xylose, see below). We found that the average community diversity increased over time, but the rate slowed as the community matured (Fig. 2b; note the logarithmic *x*-axis scale). Communities displayed significant variability in the trajectories of

increasing diversity (Fig. 2b, gray lines). We discuss the slow increase of diversity, and observed variability, in the next section.

*Top choice complementarity.* The top choice complementarity of a community measured the overlap in the top choice resource of each of the species residing in the community. We defined the top choice complementarity of a community as the number of unique top choice resources among community residents, divided by the number of residents. Thus the top choice complementarity varied between 1, in a maximally complementary community where each resident species had a unique top choice resource (Fig. 2c, right), and 1 divided by the number of coexisting species in the community, where all residents chose the same resource as the top choice (Fig. 2c, left). During community assembly, top choice

complementarity stayed close to the maximum value throughout the assembly process (Fig. 2c, blue). This observation was in sharp contrast to the prediction from a null model for the complementarity (Fig. 2c, red). We obtained the null prediction by measuring the complementarity of a group of randomly chosen species from the species pool (group size being the number of coexisting species in the community). This null prediction decreased during the assembly process, due to the increasing community diversity, unlike the top choice complementarity which remained close to the maximum value. We also recorded the complementarity in the second, third, and fourth choice resource of the assembled community (defined similarly to the top choice complementarity). The complementarity of all other choices agreed with the null prediction (Fig. 2c). Together, these observations suggest that communities of coexisting diauxic species exhibit high complementarity on the top-choice resources, in a manner reminiscent of niche partitioning in consumer-resource models.

*Prevalence of anomalous species.* Intuition gleaned from experiments with *E. coli* dictates that microbes often grow fastest on their top choice resource (glucose for *E. coli*)[18,20]. However, exceptions to this trend also exist, such as Bacteroides species in the human gut that often prefer polysaccharides that they grow slower on[22,26,38]. Based on this intuition, we defined *anomalous* microbes as microbes that do not grow fastest on their top choice resource. To investigate which resource preferences might give microbes a competitive advantage during community assembly, we tracked the fraction of anomalous resident species during community assembly. Despite the majority (75%) of species in the pool being anomalous (since growth rates and preferences were randomly picked; see "Methods"), anomalous species were absent in mature communities. The fraction of anomalous resident species decreased rapidly during assembly (Fig. 2d). Thus, anomalous resource preferences are strongly selected against during community assembly. Further investigation revealed a reduced selection pressure against anomalous species if either resource supply was severely imbalanced (i.e., the imbalance has to be comparable to the dilution factor, $D = 100$), or if the dilution factor was small (see Figs. S4 and S5; also see Supplementary Text, sections C and H). However, microbes with anomalous resource preferences were eventually outcompeted in all conditions.

**Top choice resources chiefly drive emergent assembly patterns**. To understand what factors drove the maintenance of top choice complementarity—despite the steady increase in species diversity, expected to reduce complementarity—we focused on growth on top choice resources. We hypothesized that the reason for the much higher than expected top choice complementarity was the following: diauxic species derived most of their growth, and spent most of their time growing on their top choice resources. Co-utilizing microbes, instead, grow on multiple resources simultaneously, spending roughly equal time on each utilized resource.

To test this hypothesis, we first simulated the growth of a single diauxic species in monoculture using our model. We found that indeed, the species derived the overwhelming majority of its growth (measured in generations of growth) and spent most of its time growing on its top choice resource (54%, Fig. 3a, b, left). For a simpler case, where a single species had the same growth rate $g$ while growing on two resources (both supplied at the same concentration), and preferring resource $R_1$ over $R_2$, we derived the ratio of time spent growing on the top choice resource $R_1$ ($T_1$) versus the second choice $R_2$ ($T_2 - T_1$). We obtained the following approximate expression for a large dilution factor $D$ (see

Supplementary Text, section A):

$$\frac{T_1}{T_2 - T_1} = \frac{\log(D/2)}{\log(2)}, \qquad (1)$$

which explains that the fraction of time spent growing on the top choice resource increases with the dilution factor.

Strikingly, the fraction of time spent growing on the top choice resource became even larger if the species grown in monoculture (Fig. 3b, top row) were instead part of a diverse community (i.e., in co-culture with 3 other species, top choice share 70% versus 54% in monoculture, Fig. 3b, bottom row and top row, respectively). This is because of the following reason. In our model, while a species consumes and grows on all available resources in monoculture, in co-culture, it may not have the opportunity to consume all the resources it can grow on because other species might deplete them first. This further skews growth in favor of the top choice resource. Such a phenomenon only occurs in diauxic species, not co-utilizing species (Supplementary Text, section I).

*Invader success.* Interestingly, once we understood that the top choice chiefly drove species growth, we could explain the other emergent patterns in diauxic communities. Importantly, the success of an invader depended on the growth rate on their top choice resource. As community assembly proceeded, the top choice growth rate of successful invaders increased consistently (Fig. 3c, blue line), while their growth rates on all other choices remained constant and close to the average growth rate (Fig. 3c, green lines). Selection on the top choice growth rate in diauxic communities is in striking contrast with co-utilizing communities, which we found select for the average growth rate across all resources instead (Supplementary Text, section I). Further, an invader whose top choice resource coincided with the last depleted resource in the community had the highest probability of invasion success (Fig. 3d). Invaders whose top choice resource was not depleted last had lesser time to grow on it, and thus a lower rate of invasion success. By depleting the last resource faster, invaders reduced the total resource depletion time in the community, thus explaining the trend observed in Fig. 2a. In addition, after a successful invasion, the community's steady state could have a different resource depletion order.

*Complementarity and diversity.* Successful invasions could be classified into one of two types based on the "invaded resource", i.e., the invader's top choice. If the invaded resource was not the top choice of any other resident community member, we called it an invasion of an "unoccupied" resource (Fig. 3e; in our simulations, 33% of cases). If the invaded resource was instead already the top choice of at least one resident, we called it an invasion of an "occupied" resource (Fig. 3e; 67% of cases). Both types of successful invasions had different effects on species diversity, but interestingly, both maintained complementarity (on the top choice, as in Fig. 2c). invasions of unoccupied resources usually increased community diversity by 1 (62% of cases), and were less likely to result in the extinction of one or more other species (38% of cases). This is because, in that case, the invader did not have to compete with other residents for its top choice resource. For communities with a complementarity <1, unoccupied invasions increased complementarity (Fig. 3e). For communities with complementarity 1, they maintained complementarity (Fig. 3e).

In contrast, an invasion of an occupied resource usually brought the invader in conflict with at least one resident with the same top choice (Fig. 3f, cartoon). To be successful, an invader typically had to have a better growth rate than the resident with a matching top choice. A successful invasion of this kind thus typically maintained species diversity (by replacing one resident with the invader;

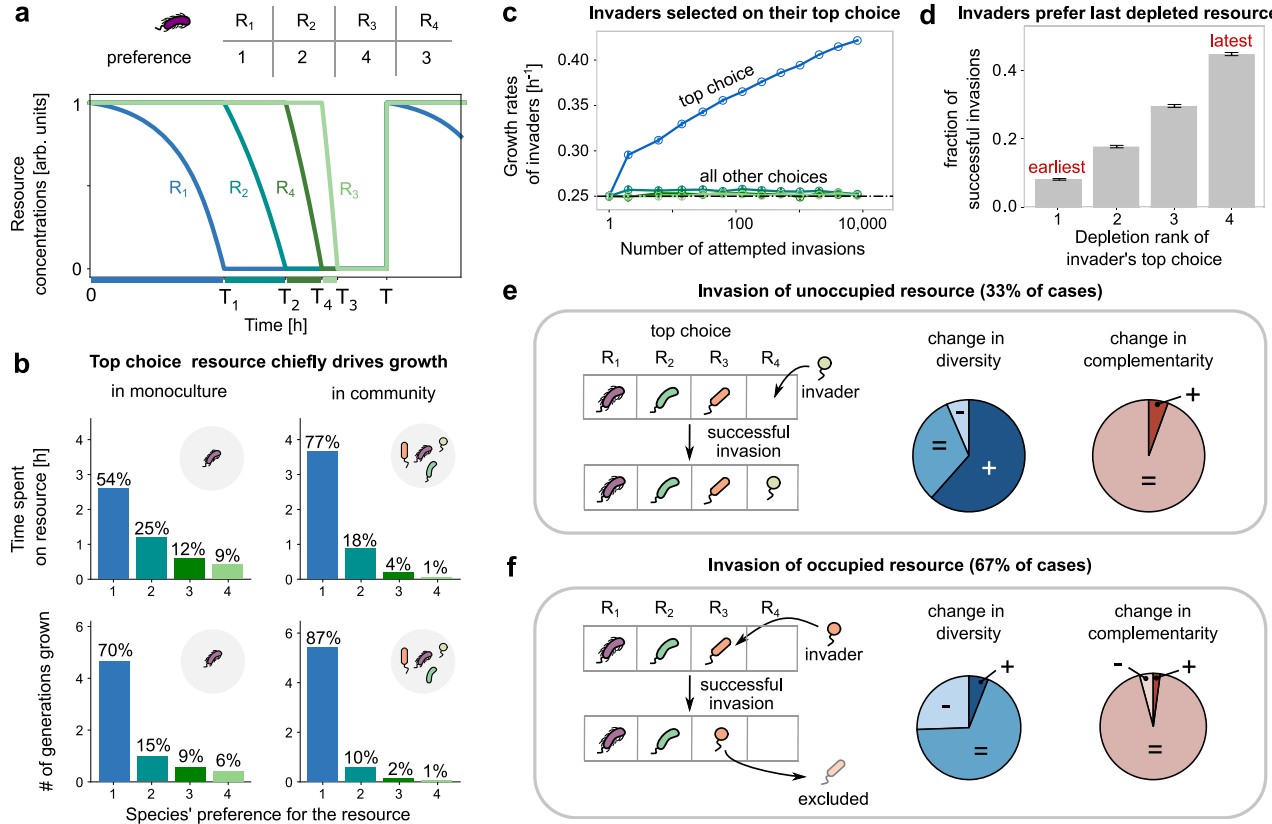

**Fig. 3 Top choice resources chiefly drive community diversity and complementarity. a** (top) Table showing the preferences of a diauxic microbial species (purple) for 4 resources, $R_1$ to $R_4$. (bottom) Plots showing the depletion of the 4 resources by the purple species during one serial dilution cycle, when grown alone in our model. **b** (top) Bar plots showing the time taken by the purple species in (**a**) to grow on each of the 4 resources. Percentages on each bar represent the fraction of time spent growing on each resource. (bottom) Bar plots showing the number of generations grown, or the number of doublings by the species when growing on each resource. In both cases, the plots on the left show the quantities when the purple species is in monoculture (growing alone), and those on the right show them when the purple species is in a community with 3 other species. **c** Mean growth rates of successful invaders during community assembly. The blue line corresponds to the invader's top choice, while the other colors correspond to all other choices. The horizontal dashed line shows the mean growth rate of the species pool. Each quantity represents a moving average from 958 independent community assembly simulations. Error bars represent s.e.m. **d** Fraction of the successful invasions as a function of the order in which the invader's top choice resource is depleted, 1 indicating cases where the invader prefers the earliest depleted resource, and 4 where it prefers the last depleted resource. Each bar represents the mean of such a fraction over 958 independent community assembly simulations, and error bars represent s.e.m. **e, f** Effect of invasions of community diversity and complementarity, based on whether the invader's top choice was (**e**) unoccupied or (**f**) occupied. Cartoons show the typical effect of an invasion. Pie charts show the fraction of invasions that increase, decrease or maintain a community's species diversity (middle) and complementarity (right). On unoccupied resources, diversity typically increases (62%), but sometimes stays constant (32%) or decreases (6%). On occupied resources, diversity typically stays the same (68%), but sometimes decreases (26%) and rarely increases (6%). In almost all cases complementarity either stays maintained or increases (>95%), and very rarely decreases (<5%). Source data are provided as a Source data file.

Fig. 3f) and sometimes, could decrease it (by knocking out more species; Fig. 3f). It is because of the prevalence of invasions of this kind (94% of invasions)—which often did not change or could even decrease diversity—that overall species diversity increased slowly in the community. However, because the outcome of all such invasions was driven primarily by the top choice resources, replacements by successful invaders maintained top choice complementarity (Fig. 3f). Thus, even though successful invasions could impact species diversity in different and often contrasting ways, they maintained species complementarity on their top choices. Moreover, because all other choices had much smaller effects on species' growth, the emergence of complementarity was restricted to only the top choice.

**A geometric approach to understand microbial communities undergoing serial dilution**. To further understand what shaped the observed emergent community properties, we developed a geometric approach to visualize, analyze, and understand community assembly during serial dilution. The approach is inspired by previous work by Tilman[39,40], where he developed geometric methods to analyze continuously diluted (chemostat-like) communities. The geometric method is easiest to visualize for a community in a two-resource environment, and so we will restrict ourselves to this scenario in this text.

The key insight to developing a geometric approach for serially diluted communities is the following: each resource environment can be characterized by a set of steady-state resource depletion times, $T_i$ in our model. At steady state, a species starts consecutive growth cycles at the same abundance i.e., its abundance grows by a factor equal to the dilution factor $D$ every growth cycle. The set of resource depletion times that allows a species to grow exactly by a factor $D$ defines a set of curves in the space of $T_i$. We term these curves zero net growth isoclines (ZNGIs) following Tilman and others[39–43].

Figure 4a illustrates the ZNGI for a species $\alpha$ growing on two resources, $R_1$ and $R_2$), preferring $R_2$ over $R_1$. The isoclines are

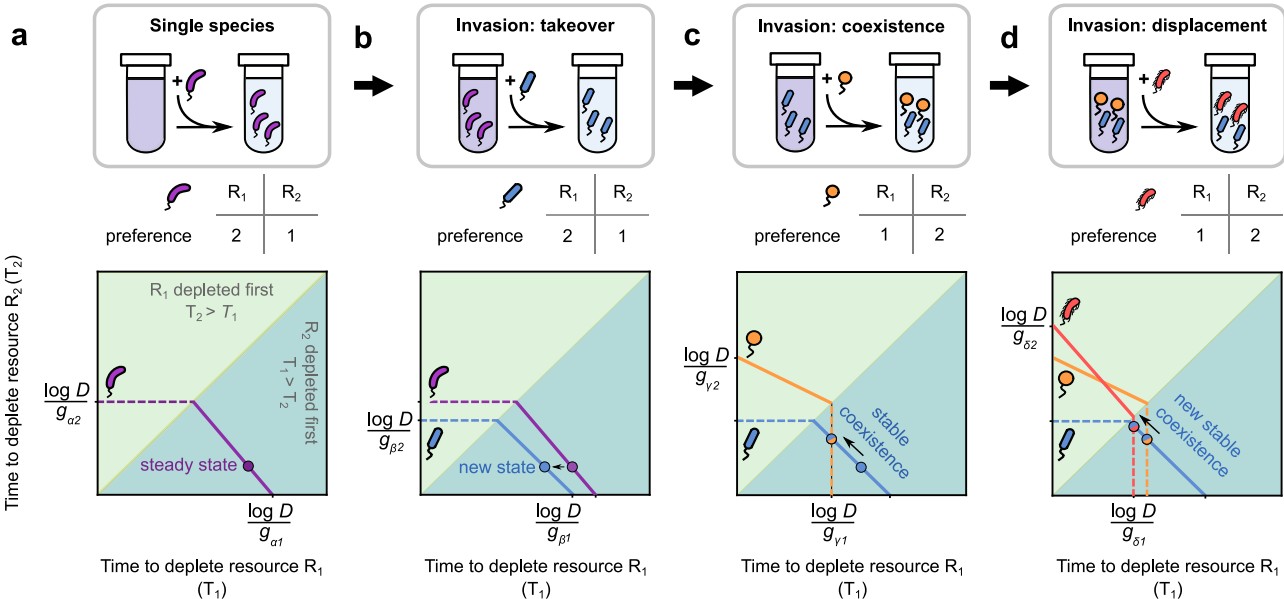

**Fig. 4 A geometric approach for serially diluted communities explains how complementarity promotes diversity.** Representation of stepwise community assembly in our geometric approach. **a** Resource utilization time plane for a single diauxic species (purple) growing on 2 resources, $R_1$ and $R_2$. The x-axis represents the time taken to deplete $R_1$, and the y-axis for $R_2$. Each species is characterized by a zero net growth isocline (ZNGI) comprising two line segments: a dashed line indicates the less preferred resource, while the solid line indicates the more preferred resource (the table shows preferences). In monoculture, the steady state can only occur on the solid line, here the bold circle. The $T_1 = T_2$ line separates the plane into two triangular regions: in the light green region, $R_1$ is depleted first, while in the dark green, $R_2$ is. **b** Representation of a new species invasion, where the invader (blue) takes over and displaces the purple resident. The invader has the same resource preferences as the resident, but its ZNGI is closer to the origin, suggesting that it depletes resources faster. The new steady state is shown as a bold blue circle. **c** Representation of a new species invasion, where the invader (orange) coexists with the new resident (blue). The invader has complementary preferences to the resident, indicated by the solid segment of its ZNGI in the light green region. Both species stably coexist (see Supplementary Text, sections A and B) at the intersection of their ZNGIs (orange/blue circle). **d** Representation of a new species invasion, where the invader (red) displaces only one of the residents, that with the same top choice resource (orange). The invader (red) and resident (blue) coexist stably at the intersection of their ZNGIs (red/blue bold point). In all cases, the point marking the steady state moves closer to the $T_1 = T_2$ line, showing that the depletion times for both resources become similar to each other.

composed of two separate lines in the two triangular regions: $T_1 < T_2$ (Fig. 4a, yellow) and $T_1 > T_2$ (Fig. 4a, green) defined by the following equations:

$$g_{\alpha 2} T_2 = \log D, \quad \text{if } T_1 < T_2$$
$$g_{\alpha 1}(T_1 - T_2) + g_{\alpha 2} T_2 = \log D, \quad \text{if } T_1 > T_2, \quad (2)$$

where $g_{\alpha 1}$ and $g_{\alpha 2}$ are the species $\alpha$'s growth rates on $R_1$ and $R_2$, respectively. The two triangular regions separated by the diagonal define two complementary scenarios: when $T_1 < T_2$, $R_1$ is depleted first and the species grows exclusively on its preferred resource $R_2$; when $T_1 > T_2$, $R_1$ is depleted second and the species grows on $R_1$ after $R_2$ is depleted.

For a given set of initial resource and species concentrations, community dynamics must converge to a steady state lying on the ZNGI of the surviving species (e.g., the bold purple point in Fig. 4a). This point defines the resource depletion times by the resident species at steady state. Changing the resource supply or dilution factor moves this point along the ZNGI.

The ZNGI of a species also separates the resource environment space into two regions: a region inside the ZNGI (towards the origin) where that species grows by a factor $<D$, and a region outside the ZNGI (away from the origin) where the species grows by a factor $>D$. An invader is successful if it is able to grow by a factor $\geq D$ in the community it invades. Geometrically, the invader's ZNGI must be closer to the origin than the resource environment corresponding to the invaded community (Fig. 4b). In this way, our geometric approach allows easy visualization of invasion criteria.

We can also visualize invasion outcomes. A successful invasion of a single-species community leads to either displacement of the resident or coexistence between the invader and resident. For example, in Fig. 4b, because the ZNGI of the invader (blue) lies fully inside the ZNGI of the resident (purple), the invader displaces the resident. This is because the invader reduces the resource depletion times in the environment to a point where the resident can no longer survive, driving it extinct (bold blue point in Fig. 4b). In contrast, in Fig. 4c, the ZNGI of the invader (orange) intersects with the new resident (blue), in a manner that leads to coexistence between both species (albeit at a new set of resource depletion times, i.e., their intersection point in Fig. 4c). In general, whether two species will coexist depends on various factors, such as the supplied resource concentrations, but whenever two species coexist, they will do so at the intersection of their ZNGIs (Supplementary Text, section A). As a corollary, two species whose ZNGIs do not intersect cannot coexist. Notably, the orange and blue species in Fig. 4c coexist stably with each other; a short perturbation to the resource supply is quickly compensated by species growth, and the resource depletion times returned to the coexistence point (see Supplementary Text, section B for details).

The geometric approach provides an alternative explanation to why species with complementary top choices are more likely to coexist than species with the same top choice (Fig. 2c). The ZNGIs of species sharing the same top choice are unlikely to intersect with each other (e.g., the blue and purple species in Fig. 4b). This is because of two reasons: (1) their segments in the yellow region are parallel to each other since both species prefer

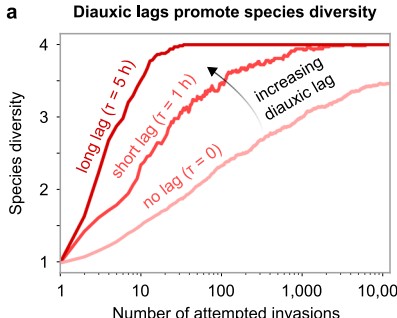
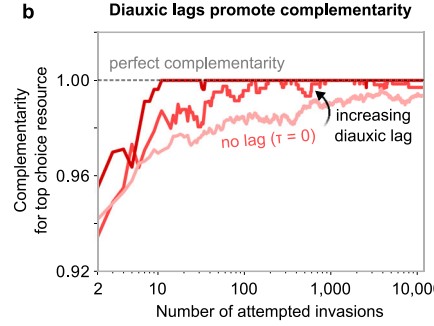

**Fig. 5 Diauxic lags further strengthen complementarity for top choice resources. a** Species diversity of communities during community assembly, averaged over 100 independent community assembly simulations of our model. The plot shows three cases: with no diauxic lags $\tau = 0$ (pink), with a short lag $\tau = 1\,h$ for all species (light red), and with a long lag $\tau = 5\,h$ for all species (dark red) between all resources ("Methods"). **b** Complementarity for the top choice resource during community assembly, averaged over the same simulations in (**a**). The horizontal dashed line highlights perfect complementarity, i.e., 1.

the same resource ($R_2$), and (2) for the slanted segments in the green region to intersect, the blue species would need a higher growth rate on $R_1$ than the purple species. This is as likely as the outcome of a coin toss, since both growth rates derive from the same distribution. Thus, an invasion of an occupied resource often leads to displacement of the resident, not coexistence (Fig. 4b, d) and no change in community diversity, while an invasion of an unoccupied resource often leads to coexistence (Fig. 4c) and an increase in community diversity (Fig. 3e).

Finally, we can also explain the relative scarcity of anomalous species (Fig. 2d). The ZNGI of an anomalous species is defined by an acute angle between the two line segments that comprise it (Supplementary Text, section C and Fig. S11), in contrast with an obtuse angle for a non-anomalous species (e.g., Fig. 4a). Briefly, it is possible for an anomalous species to displace a species with the same resource preference, but nevertheless be displaced by that same species once community diversity increases. This scenario is described in detail in the Supplementary Text, section C. Taken together, our geometric approach helps explain all the emergent properties of diauxic communities described in the text.

**Diauxic lags promote complementarity, coexistence, and multistability.** In our model, we have so far assumed the absence of diauxic lags to simplify our calculations and simulations ($\tau = 0$). Diauxic lags represent a period of no growth after exhausting a resource during which microbes re-wire their metabolic machinery for utilization of the next resource[27,30,44,45]. To test how adding diauxic lags would affect our results, we considered two variants of our model, first with fixed lag times, and second with variable lag times in an experimentally relevant scenario.

In the first variant of our model with lags, we added a fixed diauxic lag uniformly for all species and resources, and repeated our community assembly simulations. We found that adding diauxic lags promotes species diversity as well as complementarity, thus strengthening our results (Fig. 5; Supplementary Text, section H). As microbes spend more time switching from one resource to another, they wait longer to grow on less preferred resources. Increasing lag times thus skews their growth even more towards their top choice resource. Recent experimental work has shown that there is a trade-off between diauxic lag times and microbial growth rates, such that microbes that grow faster tend to experience longer lags[30]. Adding such a growth-lag trade-off to our model promotes diversity and complementarity even further (Figs. S3 and S6). Our geometric approach explains why a growth-lag trade-off promotes coexistence. Namely, there is a greater chance for the ZNGIs of two microbes exhibiting such a trade-off to intersect with each other and lead to stable coexistence (see Supplementary Text, section H).

In the second variant, we applied our model to predict the outcomes of pairwise co-cultures of 13 laboratory *E. coli* strains. We knew the growth parameters, such as growth rates and yields, for these strains from experimental measurements (data from Barthe et al.[46]). These strains represented a much less diverse pool than the one we had assumed so far, and thus all had the same preference order, preferring glucose over xylose (Fig. 6b). While the diauxic lag times were measured in a population shifting from glucose to xylose for the first time, they were shown to be sensitive to experimental conditions and characterized by long memory effects, and thus could potentially vary across multiple dilution cycles. Further, the initial (regrowth) times of these strains relevant for the serial dilution experiments were not measured. Therefore, we sampled both the diauxic and initial lag times from uniform distributions in the reported experimentally relevant ranges (Fig. 6c; see "Methods"). For a randomly chosen set of lag times, our simulations yielded three interesting observations. First, even for these closely related strains, there was a wide variety of qualitative outcomes (Fig. 6d), ranging from competitive exclusion (95%), coexistence (4%), and in rare cases, bistability (where both species mutually excluded each other; 1%). Notably, these results show that bistability is another possible outcome for diauxic species (see Supplementary Text, section G for a more detailed discussion). In our simulations, bistability resulted from a difference in biomass yields for glucose and xylose across strains. Using geometric theory developed in our study, we found that bistability (and even multistability) can also result from other factors, such as the presence of diauxic lags, and anomalous species (Supplementary Text, section D and H). Second, strains with the same preference order could coexist (Fig. 6e), showing that competition between strains did not only occur on the top choice, but on multiple resources. A strain that grows slower on its top choice (i.e., an anomalous species) could still coexist if it grew faster on its second choice, or if it had a shorter lag (Fig. S22). Third, the qualitative co-culture outcomes were relatively robust to varying the lag (both initial and diauxic) times, suggesting that the uncertainty in these parameters was unlikely to affect most qualitative outcomes (Fig. 6e). Multiple simulations where we randomly sampled the lag times in an experimentally relevant range showed that most outcomes (e.g., coexistence) were robust to uncertainty in the lag times. Together, these results show that the presence of lags strengthens our results, ultimately allowing us to make testable predictions for laboratory strains.

## Discussion

The vast majority of resource-explicit models used to describe microbial community assembly assume that microbes simultaneously co-utilize multiple resources[11,13,17,33,47]. However, ever

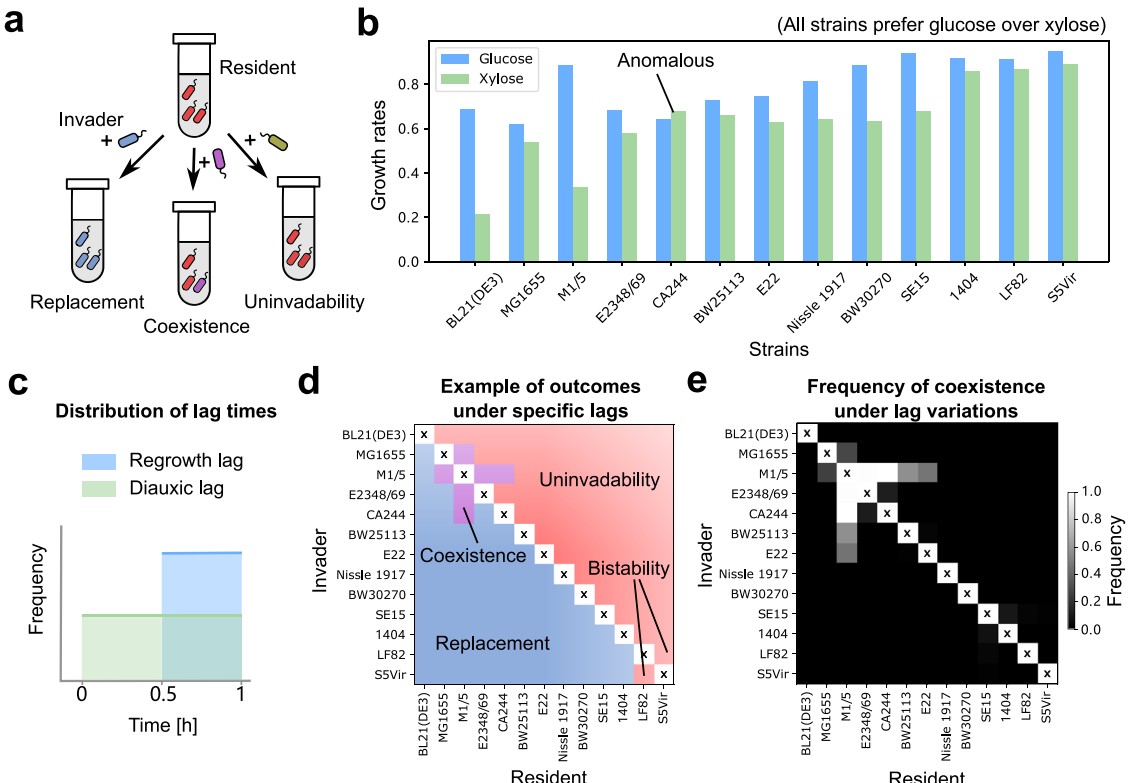

**Fig. 6 Predicting pairwise co-culture outcomes of diauxic laboratory *E. coli* strains. a** Schematic of pairwise invasion simulations. For each invasion event, a small amount of an invader strain was added into the monoculture of the resident strain. The system was then serially diluted until reaching a steady state. We simulated these pairwise invasions using *E. coli* strains discussed in ref. [46], where their growth rates and yields were measured in experiments. An invader either replaces the resident, coexists with it, or is unable to invade. **b** Growth rates of the 13 laboratory *E. coli* strains on glucose (blue) and xylose (green), obtained from Barthe et al.[46]. Note that one of the strains is anomalous. **c** Distribution of the initial lags (uniform between 0.5 and 1 h) and diauxic lags (uniform between 0 and 1 h) used in simulations. In each of 100 simulation runs, we randomly sampled the lag times of each strain, and performed exhaustive pairwise invasion simulations. **d** An example of competitive outcomes, showing 1 of the 100 runs. In most circumstances, the invader beats the resident (blue) or fails to invade (red). Coexistence (purple) or bistability can also occasionally happen. **e** Frequency of coexistence. A strain on the y-axis invades the monoculture of a strain on the x-axis, and the greyscale represents the frequency that both strains survive in the steady state among 100 runs where lag times were sampled.

since the 1949 paper by Jacques Monod it has been known that many microbes utilize resources sequentially via diauxie, not simultaneously[18]. Here, by incorporating sequential resource utilization as the dominant mode of microbial growth, we have developed a minimal model of communities undergoing diauxic shifts. Using the model, we have discovered several emergent properties of stochastically assembled communities. The most striking of these properties is a natural tendency for all surviving community members to prefer different and unique top resources (i.e., the resource a member consumes first), nearly perfectly complementing each other. In our model, communities maintain such perfect complementarity despite significant perturbations that arise due to the assembly process, such as the invasion of new species and secondary extinctions.

We also identified the key reason that top choice complementarity spontaneously emerges in diauxic microbial communities under feast-and-famine cycles, which serial dilutions mimic. That is, the top choice resource contributes overwhelmingly to microbial growth compared with all other resources that a microbe can use during each dilution cycle. This asymmetric preference towards the top resource means that it is highly unlikely (though possible) for species with the same top resource to coexist in a multispecies community. In this way, top choice complementarity naturally emerges in fully diauxic microbial communities, purely through community assembly. Such complementarity has indeed been observed in natural

communities, but only a few cases have thus far been explored[26,29,48,49]. Our results provide a clear and testable prediction for future experiments: synthetic communities composed of several diauxic shifting microbes should reach a steady state with survivors that have nearly perfect top-choice complementarity.

Our minimal model with diauxic shifts also captures many features of microbial communities, such as multistability, priority effects, and non-transitive (e.g., rock-paper-scissors) interactions (Supplementary Text, sections D and E). Notably, while these features are either absent or rare in models with co-utilization, they emerge naturally in our model. This is in part due to the discrete nature of growth that diauxie enables, which our geometric model exploits to explain their emergence. While a full discussion and exploration of these phenomena are outside the scope of the main text, we have discussed their existence in some detail in the Supplementary Text. Since these features, as well as diauxic shifts, are in fact present in real communities, a more thorough exploration of them remains an outstanding goal for future work.

For simplicity we assumed a balanced resource supply, i.e., we supplied every resource at the same concentration. By performing simulations with a few unbalanced resource supplies, we found that providing a few resources at a much higher concentration than others reduces complementarity, but only when the imbalance between resource supplies is extremely large (comparable to

the dilution factor, $D \approx 100$; see Supplementary Text, section C and Fig. S5). Thus, even outside the parameters and simplifications presented in the main text, we expect that complementarity should be prevalent in diauxic communities.

In our model, we found that anomalous species always went extinct, but anomalous microbes are indeed found in nature, albeit rarely. Examples are certain *Bacteroides* strains in the human gut microbiome[26]. One way to promote the survival of anomalous microbes in our model is to add diauxic lags, as well as a "super" resource, whose mean growth rate is significantly higher (140%) than other resources. As the diauxic lag times increase, so does the fraction of anomalous microbes that survive (~5–10% with a 1 h lag; Fig. S8). In this scenario, a non-anomalous survivor would tend to use the super resource as its top choice. An anomalous species that prefers another resource can completely deplete it and thereby survive if the other species experiences a long enough diauxic lag before switching resources (Fig. S8, 1 h is enough).

Our observation that the top choice resource contributes overwhelmingly to diauxic growth might suggest that species should tend to lose their ability to consume other resources, instead of becoming specialists on their top choice. However, resource availability fluctuates in the environment, and would prevent species from becoming specialists. Importantly, the complementarity of any community strongly depends on which environmental resources are available. The top choice of any diauxic microbe is the resource which it grows on first in a given resource environment. If the available resources were to change, so would a species' top choice, in turn becoming the top resource in its list of preferences among the ones available. Thus, when environments fluctuate, even in a single microbial species, complementarity would select for growth rates on multiple resources, not just one unique resource. In this way, species would tend to maintain the ability to use multiple resources, not just a singular top choice resource.

Alongside our model and simulations, we presented a geometric framework for interpreting our results inspired by work from Tilman and others[39,40]. This framework explains how resource depletion times, not resource concentrations, are more crucial for determining community states when undergoing serial dilution (see Fig. 4 for diauxic communities; see Supplementary Text, section I for co-utilizing communities). The geometric method also intuitively explains how complementarity affects coexistence, and why species with anomalous resource preferences can be competitively excluded from diauxic communities. The framework thus provides a rich theoretical resource not just to understand diauxic communities, but generally to any communities undergoing discrete mortality events (feast-famine cycles) or dilutions. Given the large number of community assembly experiments in laboratories are performed using such a protocol, we believe that it has the potential to generate many more experimentally testable hypotheses.

**Testable predictions.** Finally, we provide a set of testable predictions that result from this work. As described above, our chief prediction is as follows: steady-state laboratory communities of diauxically shifting microbes should be mainly composed of microbes which prefer unique top choice resources (Fig. 5b). We have three additional predictions, which concern invasions in microbial communities. First, an invader should more successfully establish in communities which are slow to deplete its top choice resource (Fig. 3d). Second, if an invader successfully establishes in a community, resident species with the same top choice as the invader would likely get excluded from the community (Fig. 3f). Third, if a successful invader is an anomalous

species, i.e., it grows fastest on a resource other than its top choice, then it is likely to have other compensatory mechanisms to establish in a community (Fig. 2d). Consistent with this prediction, recent experimental work has shown that certain *Pseudomonas* species which exhibit unusual carbon source preferences (reverse diauxie) utilize virulence genes to establish in communities[50]. Finally, using experimentally measured growth rates and lag times, our model predicted the competitive outcomes of all pairwise co-cultures of 13 laboratory *E. coli* strains, which can be tested experimentally (Fig. 6). These predictions improve our understanding of the effect of invasions of microbial communities, and demonstrate the value of studying diauxic community assembly.

## Methods

**Generating species pools.** To simulate the community assembly using our model, we generated a set of 958 independent species pools. As described in the "Results" section, we assumed that all communities were supplied with 4 unique resources. Since each microbial species was diauxic, there were a total of 24 possible resource preference orders that a species could have. While generating each species pool, we generated an equal number of species (480) with each unique preference order, resulting in a total of 11,520 unique species in each pool. Each species could consume all 4 resources, but with different growth rates, sampled independently from a normal distribution with a mean $0.25\,h^{-1}$ and standard deviation $0.05\,h^{-1}$, truncated below zero to ensure that growth rates were always positive. When concerning unequal growth rates distributions, we varied the mean of 1 resource's growth rates (the "super" resource), and kept that of the rest 3 resources unchanged. The variance of all growth rates also remained the same. We assumed that the yields of every species $Y$ on each resource was equal and set to 0.5.

**Model simulation.** The consumer-resource dynamics in our model mimic those of the MacArthur model[11,13], except that each species only consumes resources one at a time, sequentially. We denote each species by Greek symbols, such that the abundance of species $\alpha$ is represented by $N_\alpha$; the concentration of resources (denoted by $i$) is represented by $R_i$. We encoded diauxie through a consumption matrix, $c_{\alpha i}(t)$, which is binary and time-dependent. If species $\alpha$ consumes resource $i$ at time $t$, we set $c_{\alpha i}(t)$ to 1; otherwise, we set it to 0. For simplicity, we set the lag time between switching resources, $\tau = 0$ in our simulations. While consuming a resource, we assume that the available resource concentrations are higher than species' half-saturating substrate concentrations, such that each species grows at a constant growth rate, $g_{\alpha i}$ (which we sampled for each species while generating the species pool). This assumption is reasonable for the feast and famine scenario that we model here. Thus, species abundance dynamics in our model can be written as follows:

$$\frac{dN_\alpha}{dt} = \sum_i g_{\alpha i} c_{\alpha i}(t) N_\alpha(t). \tag{3}$$

Similarly, the resource dynamics can be written as follows:

$$\frac{dR_i}{dt} = -\sum_\alpha g_{\alpha i} c_{\alpha i}(t) \frac{N_\alpha(t)}{Y}. \tag{4}$$

To simulate the stepwise community assembly process, we introduced each species from the species pool one by one in a randomly generated order. After introducing all 11,520 species once, we introduced each unsuccessful species one by one again, until no more successful invasions were possible, i.e., the community reached an uninvadable stable state.

At the beginning of each serial dilution cycle, all 4 resources were supplied at equal concentrations, i.e., 1 unit. Each species was introduced as an invader, at a small abundance $10^{-6}$, much smaller than the abundance of any resident species in the community. During the first cycle, any invading species could grow a tiny amount, and thus have a negligible effect on the depletion times of any resources. Therefore, the invader (say species $\alpha$) would grow by a factor $\sum_{i=1}^{4} g_{\alpha i}(T_i - T_{i-1})$ in the first cycle, where $T_0 = 0$, and $T_i - T_{i-1}$ is the timespan where the species $\alpha$ grows on resource $i$, determined by resident community's resource depletion times. The criterion that an invasion succeeds is if the invader can outgrow dilution in its first cycle, i.e.,

$$N_\alpha(T)/N_\alpha(0) > D. \tag{5}$$

$D = 100$ is the dilution factor. If, for an invader species, this criterion was not met, then it did not invade, and the next invader was introduced. Otherwise, the invader was successful, and we would simulate community dynamics until a steady state. In steady state, every community member grew by a factor $D$ during a dilution cycle.

**Pairwise invasions.** To simulate pairwise invasions of some realistic examples, we took 13 specific *E. coli* strains, whose diauxic growth on glucose and xylose were experimentally measured[46].

In these experiments, initial lags, defined as the time needed to resume growth after starvation when the substrate again becomes available, were present, but were not explicitly measured. We assumed the initial lag, individually for each strain, ranges between 0.5 and 1 h. We also sampled the diauxic lags, which is the time it takes for a species to switch from consuming glucose to xylose, once glucose is depleted, from a uniform distribution between 0 and 1 h. Other parameters, such as the growth rates and yields, were obtained from the experimental results, where the strains grew on a medium of 12.5% glucose and 87.5% xylose[46]. We used the same ratio in our simulations.

In the pairwise invasion simulations, the "resident" strain was first grown in monoculture till steady state. Then the "invader" species were added at a relatively small abundance of $10^{-6}$, and we simulated serial dilutions until steady state, at a fixed dilution factor of 1000. We performed 100 such simulations, and in each run we randomly sampled the initial and diauxic lags for each strain, and performed pairwise invasions with each pair.

**Reporting summary**. Further information on research design is available in the Nature Research Reporting Summary linked to this article.

## Data availability
All the numerical data from simulations can be found on the GitHub repository, at the following link: https://github.com/maslov-group/diauxic_assembly. No new experimental data was generated during this study. Source data are provided with this paper.

## Code availability
All code comprised of custom Python scripts, using Python v3+, along with the Numpy v1.20.1 and Scipy v1.6.2 packages, and can be found at https://github.com/maslov-group/diauxic_assembly.

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

## Acknowledgements
A.G. is supported by the Gordon and Betty Moore Foundation as a Physics of Living Systems Fellow through grant number GBMF4513.

## Author contributions
S.M. and A.G. devised and supervised the study; Z.W. performed simulations and calculations. Y.F. developed and analyzed the geometric approach; Z.W., A.G., and V.D. helped with presentation and visualizations; Z.W., A.G., V.D., and A.B.G. performed data analysis; T.W. performed bistability calculations; A.G. conceptualized and wrote the paper with help from all authors.

## Competing interests
The authors declare no competing interests.
