## [Peer Review File · Nature Communications]

Reviewers' Comments:

Reviewer #1:

Remarks to the Author:

The paper by Zihan Wang and coauthors is an elegant work, clearly written and well illustrated. Starting from a very simple mathematical model describing growth of species during the assembly of diauxic microbial communities and simplifying assumptions, the authors manage to identify nice emergent properties in the assembly of microbial communities. Compared to previous works in the field, they analyse the assembly process with a different perspective, the sequential utilization of resources rather than co-utilization.

I sometimes missed biological background that would justify the modelling assumptions and give more biological impact to the work. While some of them are discussed later (e.g. the lag time), one that is not questioned is the assumption of competitive exclusion mechanism. I detail my questions related to this issue below.

- To which extent is this assumption realistic? Do the authors have some specific examples in mind that they could describe (and earlier in the text, not at the time of the discussion)?
- Restricting the assembly of a community to "competitive" species only would limit a lot the number of communities that can be built in a synthetic application. Is this a realistic scenario? Did the authors study alternative ones relaxing this assumption?
- I might have missed something but for me, the observation of a perfect top choice complementarity in Figure 2 is a direct consequence of assuming a competitive exclusion mechanism. What could have been the alternative?
- In the same line: why was the top choice complementarity higher than expected (l. 252)?
- Figures 2 and 3: I expect that the competitive mechanism necessarily introduces a selection and enriches the community with invaders that grow faster and outcompete other community members growing on the same resource at lower growth rates. Am I wrong? Could this suffice to explain the interesting result that microbes with anomalous resource preferences are eventually outcompeted in the model predictions? Which hypotheses should be relaxed in order to reproduce growth of anomalous species such as *Bacteroides* species in the human gut?

Minor comment:

- Some terminology would require a clear definition, for instance, what is called the steady state of an invader ? Of the community ?

Reviewer #2:

Remarks to the Author:

In the manuscript, Wang et al. develop a model for communities of diauxically-growing microbes, and study how they assemble in serially diluted cultures.

I think this approach is interesting, since (as the authors point out) microbial communities are rarely modeled taking into account the fact that i) in experiments bacterial cultures are most of the times diluted at regular times and ii) microbes are capable to grow diauxically, i.e. they uptake different resources one at a time and in a precise order. This approach, although relevant for experimental applications, has also been underexplored in the past, so the timing of the manuscript is appropriate. The authors also present an interesting graphical method to represent such diauxically-growing communities.

The manuscript is generally well written and of potential interest to readers. However, there are few but major issues that I think need to be addressed by authors before considering granting publication:

1. I have issues with one of the main claims of the manuscript. The authors show that the community assembly in their model leads, after several random invasion attempts, to species with complementarity in their top choice resource. The authors claim this finding to be "striking", but I have some difficulties understanding why. Let us assume, for example, that two species have the same resource preference order. Since, as shown by the authors, the species spend the vast

majority of their time growing on their top choice resource, one would expect that these two species were mainly competing for this resource, and so in the end the species with the highest growth rate on this top choice resource would eventually outcompete the other one (also the authors state in the manuscript that “the success of an invader depended on the growth rate on their top choice resource”). I would say, therefore, that it is not surprising that, with invasions of randomly sampled species, the ones that are left have complementary top choice resources. In fact, all other species will have gone to extinction because of competitive exclusion. Therefore, if we have in general N resources in the system one would expect that in the end N species with perfect complementarity in their top choice resource would be able to coexist. I also do not understand the choice that the authors do to compute the “null expectation” of the top choice resource complementarity. In fact, by measuring the overlap of randomly selected species they are effectively computing the overlap of an increasing number of random vectors, which will of course tend to zero as the number of vectors (i.e., species) increases. I therefore invite the authors to explain more clearly why they are claiming this finding to be striking. On the other hand, what I did find surprising, even though it is not stated explicitly in the manuscript, is the fact that contrarily to the expectation that I’ve described above two species with the same resource preference order are indeed able to coexist, e.g. if the species that grows slower on the top choice resource grows faster on the second preferred one, as shown in Fig. S8 (point I). This is, I think, a very interesting result that is “lost” in the Supplementary Information. I therefore invite the authors to highlight this finding in the manuscript.

2. When talking about anomalous species (i.e., species that grow slower on their top choice resource), the authors find that in their model they are eventually always outcompeted, even if some of the assumptions that they use are relaxed (i.e., balanced resource supply). However, those anomalous species, even if not common, are consistently found in microbial communities. Is there any mechanism that the authors have thought about that could allow such species to persist in the communities? I think it would be appropriate to add even a short comment on this matter.

Other comments:

A. The geometrical approach for the study of diauxically-growing communities, and the fact that it explains how resource depletion times are more crucial than resource concentrations for the properties and dynamics of this type of systems is a novel and very interesting contribution. It would be a really nice addition to the literature on the subject.

B. In the Supplementary Information, the authors show that in a system with two resources species coexistence requires the ratio R_1/R_2 of the resources’ concentrations to fall within a certain range, leading e.g. to eqs (18) and (20). I think it would make things more clear and accessible if the authors added an order-of-magnitude estimate of these ranges, using some experimentally realistic values for the parameters involved.

C. In the Supplementary Information, the authors state that “In presence of an anomalous bug, a bistability may arise”. I think this is an important prediction of their framework, which is however absent from the manuscript. I therefore invite the authors to add this prediction to the ones they are already mentioning in the manuscript.

D. There are a few typos throughout the manuscript and the Supplementary Information. I invite the authors to check and correct them.

Reviewer #1 (Remarks to the Author):

The paper by Zihan Wang and coauthors is an elegant work, clearly written and well illustrated. Starting from a very simple mathematical model describing growth of species during the assembly of diauxic microbial communities and simplifying assumptions, the authors manage to identify nice emergent properties in the assembly of microbial communities. Compared to previous works in the field, they analyse the assembly process with a different perspective, the sequential utilization of resources rather than co-utilization.

We thank the reviewer for their comments and encouragement.

I sometimes missed biological background that would justify the modelling assumptions and give more biological impact to the work. While some of them are discussed later (e.g. the lag time), one that is not questioned is the assumption of competitive exclusion mechanism. I detail my questions related to this issue below.

- To which extent is this assumption realistic? Do the authors have some specific examples in mind that they could describe (and earlier in the text, not at the time of the discussion)?

We understand the reviewer's concern surrounding our modeling assumptions. We would like to clarify that competitive exclusion was not an assumption of our model, but a natural emergent feature of consumer-resource models. In the supplementary text (section F), we show how the phenomenon of competitive exclusion, where the number of species at any steady state cannot exceed the number of resources in the medium, arises in our model.

We thank the reviewer for raising this concern, which prompted us to clarify the emergence of competitive exclusion in the Results section of the text.

As suggested, we have also included two specific examples early in the manuscript, one with *E. coli* strains in the lab, and another with *Bacteroides* strains in the human gut.

Particularly, we have included new analyses of simulations using experimentally measured growth parameters from 13 strains of *E. coli*, which all display diauxie (Fig. 6 and last section of the Results). Our model is able to predict the outcomes of co-culturing these strains pairwise in a medium with glucose and xylose. Notably, these simulations show that our model allows strains with the same resource preferences to coexist with each other. This shows that the outcome of competition depends on many factors, not strictly on the top nutrient choices of individual microbes. We have rewritten the last section of the Results to include these analyses.

Changes to the text:

Lines 188-204, *"In the model, like in other consumer-resource models, the number of coexisting species at steady state is limited by the number of resources, 4 (Fig. 2b, dashed line). This is a natural consequence of competition for resources in our model (see Supplementary Text, section F for a derivation). Notably, species with the same resource preferences can coexist in the model, as long as the number of species is less than the number of resources (e.g., pairs of E. coli strains can coexist in media with glucose and xylose, Fig. 5d). We found that the average community diversity increased over time, but the rate slowed as the community matured (Fig. 2b; note the logarithmic x-axis scale). Communities displayed significant variability in the trajectories of increasing diversity (Fig.*

2b, grey lines). We discuss the slow increase of diversity, and observed variability, in the next section.”

Lines 58-60, “Using experimental measurements of growth parameters from 13 diauxic *E. coli* strains, we predict a variety of pairwise competitive outcomes by simulating our model.”

Lines 120-130, “A realistic example of a community captured by our model is the human gut microbiome, specifically the assembly of primary consumers (e.g., *Bacteroides* species) on the polysaccharides (e.g., starch, cellulose, and mucin) that they consume. Here, there is a significant overlap between the metabolic capabilities of the microbes, but they nevertheless coexist. These species often consume polysaccharides diauxically, and engage in resource competition. Moreover, several of these species have different resource preferences, which others have hypothesized help them coexist.”

Lines 486-567, “In our model, we have so far assumed the absence of diauxic lags to simplify our calculations and simulations ($\tau=0$). Diauxic lags represent a period of no growth after exhausting a resource during which microbes rewire their metabolic machinery for utilization of the next resource. To test how adding diauxic lags would affect our results, we considered two variants of our model, first with fixed lag times, and second with variable lag times in an experimentally relevant scenario.

In the first variant of our model with lags, we added a fixed diauxic lag uniformly for all species and resources, and repeated our community assembly simulations. We found that adding diauxic lags promotes species diversity as well as complementarity, thus strengthening our results (Fig. 5; Supplementary Text, section H). As microbes spend more time switching from one resource to another, they wait longer to grow on less preferred resources. Increasing lag times thus skews their growth even more towards their top choice resource.

Recent experimental work has shown that there is a trade-off between diauxic lag times and microbial growth rates, such that microbes that grow faster tend to experience longer lags. Adding such a growth-lag trade-off to our model promotes diversity and complementarity even further (Figs. S3 and S6). Our geometric approach explains why a growth-lag trade-off promotes coexistence. Namely, there is a greater chance for the ZNGIs of two microbes exhibiting such a trade-off to intersect with each other and lead to stable coexistence (see Supplementary Text, section H).

In the second variant, we applied our model to predict the outcomes of pairwise co-cultures of 13 laboratory *E. coli* strains. We knew the growth parameters, such as growth rates and yields, for these strains from experimental measurements (data from Barthe et. al.) These strains represented a much less diverse pool than the one we had assumed so far, and thus all had the same preference order, preferring glucose over xylose (Fig. 6b). While the diauxic lag times were measured in a population shifting from glucose to xylose for the first time, they were shown to be sensitive to experimental conditions and characterized by long memory effects, and thus could potentially vary across multiple dilution cycles. Further, the initial (regrowth) times of these strains relevant for the serial dilution experiments were not measured. Therefore, we sampled both the diauxic and initial lag times from uniform distributions in the reported experimentally relevant ranges (Fig. 6c; see Methods). For a randomly chosen set of lag times, our simulations yielded three interesting observations. First, even for these closely related strains, there was a wide variety of qualitative outcomes (Fig. 6d), ranging from competitive exclusion (95%), coexistence (4%), and in rare cases,

bistability (where both species mutually excluded each other; 1%). Notably, these results show that bistability is another possible outcome for diauxic species (see Supplementary Text, section G for a more detailed discussion). In our simulations, bistability resulted from a difference in biomass yields for glucose and xylose across strains. Using geometric theory developed in our study, we found that bistability (and even multistability) can also result from other factors, such as the presence of diauxic lags, and anomalous species (Supplementary Text, section D and H). Second, strains with the same preference order could coexist (Fig. 6e), showing that competition between strains did not only occur on the top choice, but on multiple resources. A strain that grows slower on its top choice (i.e., an anomalous species) could still coexist if it grew faster on its second choice, or if it had a shorter lag (Fig. S22). Third, the qualitative co-culture outcomes were relatively robust to varying the lag (both initial and diauxic) times, suggesting that the uncertainty in these parameters was unlikely to affect most qualitative outcomes (Fig. 6e). Multiple simulations where we randomly sampled the lag times in an experimentally relevant range showed that most outcomes (e.g., coexistence) were robust to uncertainty in the lag times. Together, these results show that the presence of lags strengthens our results, ultimately allowing us to make testable predictions for laboratory strains.”

- Restricting the assembly of a community to "competitive" species only would limit a lot the number of communities that can be built in a synthetic application. Is this a realistic scenario? Did the authors study alternative ones relaxing this assumption?

The reviewer is correct to point out that we are only modeling resource competition, not cooperation via phenomena such as cross-feeding. As discussed in the previous response, we do believe that our model applies to many realistic examples, such as the case of *E. coli* strains in the lab (studied in a new figure, Fig. 6), as well as *Bacteroides* strains in the human gut, which compete with others for polysaccharides.

That being said, cooperation and cross-feeding are indeed commonly observed in microbial communities. We did not incorporate cross-feeding to keep our model and analysis simple. As we hope the reviewer will appreciate, our model can be extended to include cross-feeding. However, doing so will introduce several new parameters, such as a matrix listing which species produce which byproducts, because of which such an extension remains outside the scope of the present manuscript. Additionally, we believe that adding cross-feeding would likely not affect our main results about complementarity. This is because in each cycle, byproducts would have to accumulate and would not be beneficial as top choices (they will have low concentration until they accumulate in appreciable concentrations). They will ideally be good second choices, in which case top choice complementarity would still be observed.

In the revised manuscript, we have highlighted examples where our model can be applied, such as in the last section of Results.

- I might have missed something but for me, the observation of a perfect top choice complementarity in Figure 2 is a direct consequence of assuming a competitive exclusion mechanism. What could have been the alternative?

The reviewer brings up an important question regarding competitive exclusion in our model. If our understanding of the reviewer's comment is correct, our original manuscript gave the impression that competitive exclusion is an assumption of our model, and occurs only based on the top choice. As we described in a previous response, this is not the case. Competitive exclusion emerges naturally in consumer-resource models (derived in Supplementary Text,

section F). Most importantly, species with the same top choice can coexist in our model, as evidenced in our model's application to *E. coli* strains from the lab, shown in the new Fig. 6. This would have not been possible if our model assumed that competition occurred solely on the top choice.

We apologize that our original manuscript did not clarify this aspect of our model. We thank the reviewer for helping us realize this. In the revised manuscript, we have explicitly discussed how competitive exclusion arises in our model, as well as the fact that it occurs based on multiple resources, not just the top choice, as evidenced by the observation that species with the same top choice can coexist.

- In the same line: why was the top choice complementarity higher than expected (l. 252)?

We measured the expected complementarity as follows: for each community, we measured the number of unique resources among the survivors' top choices, divided by the total number of survivors. To provide an average null expectation, we calculated the mean of this quantity over all 998 community replicates. This average null expectation decreases as communities assemble and become more species-rich, since the probability of at least two species sharing the same top choice by chance increases with the number of species in the community.

The top choice complementarity (line 252 in the original manuscript) was higher than expected because in our simulated communities, virtually all surviving species had different and unique top choices. This occurs as a consequence of community assembly and niche segregation, as sets of species which use different top choices are more likely to coexist and grow at steady state.

- Figures 2 and 3: I expect that the competitive mechanism necessarily introduces a selection and enriches the community with invaders that grow faster and outcompete other community members growing on the same resource at lower growth rates. Am I wrong?

The reviewer is correct, but it is important to note that exceptions also exist. As highlighted before, invaders with the same top choice as a resident species, and a better growth rate on it, can coexist with the resident species. This is because of the presence of other resources, which in occasional circumstances, allow microbes that grow slower on top choices to accumulate biomass much faster on their secondary choices, thereby allowing coexistence.

In the revised manuscript, we have performed new analysis of experimental data to strengthen this point and highlight it in a biologically relevant context.

Could this suffice to explain the interesting result that microbes with anomalous resource preferences are eventually outcompeted in the model predictions? Which hypotheses should be relaxed in order to reproduce growth of anomalous species such as *Bacteroides* species in the human gut?

This is a very pertinent question, since anomalous microbes like certain *Bacteroides* strains indeed exist in nature. We thank the reviewer for encouraging us to explore which assumptions could be relaxed to allow anomalous species to survive in the model, and not go extinct.

One way to promote the survival of anomalous microbes in our model is to add diauxic lags, as well as a "super" resource, whose mean growth rate is higher than other resources. As the

diauxic lag times increase, so does the fraction of anomalous microbes that survive (~5-10% with a 1 hr lag; Fig. S8). In this scenario, a non-anomalous survivor would tend to use the super resource as its top choice. An anomalous species that prefers another resource can completely deplete it and thereby survive if the other non-anomalous species experiences a long enough diauxic lag before switching resources (Fig. S8, 1 hr is enough; Supplementary Text, section K). As the diauxic lag times increase, so does the survival probability of anomalous microbes.

In the revised manuscript, we have included a new supplementary figure to show results from simulations where anomalous species, which satisfy the two criteria we described above, are likely to persist in our communities. We have also added new text to highlight these examples, and connect them with other mechanisms (e.g., inhibitory interactions) that may allow some anomalous microbes to persist in natural microbial communities. We also have new analytical derivations in the supplementary text that explicitly show what conditions must be satisfied for anomalous species to coexist in such a scenario.

Changes to the text:

Lines 635-650, *“In our model, we found that anomalous species always went extinct, but anomalous microbes are indeed found in nature, albeit rarely. Examples are certain Bacteroides strains in the human gut microbiome. One way to promote the survival of anomalous microbes in our model is to add diauxic lags, as well as a “`super” resource, whose mean growth rate is significantly higher (140%) than other resources. As the diauxic lag times increase, so does the fraction of anomalous microbes that survive (~5-10% with a 1 hr lag; Fig. S8). In this scenario, a non-anomalous survivor would tend to use the super resource as its top choice. An anomalous species that prefers another resource can completely deplete it and thereby survive if the other species experiences a long enough diauxic lag before switching resources (Fig. S8, 1 hr is enough).”*

Minor comment:

- Some terminology would require a clear definition, for instance, what is called the steady state of an invader ? Of the community ?

It is indeed the steady state of the community once an invader is introduced. We have now corrected this.

Reviewer #2 (Remarks to the Author):

In the manuscript, Wang et al. develop a model for communities of diauxically-growing microbes, and study how they assemble in serially diluted cultures.

I think this approach is interesting, since (as the authors point out) microbial communities are rarely modeled taking into account the fact that i) in experiments bacterial cultures are most of the times diluted at regular times and ii) microbes are capable to grow diauxically, i.e. they uptake different resources one at a time and in a precise order. This approach, although relevant for experimental applications, has also been underexplored in the past, so the timing of the manuscript is appropriate. The authors also present an interesting graphical method to represent such diauxically-growing communities.

We thank the reviewer for their rigorous reading and insightful suggestions, which we think greatly improved our manuscript.

The manuscript is generally well written and of potential interest to readers. However, there are few but major issues that I think need to be addressed by authors before considering granting publication:

1. I have issues with one of the main claims of the manuscript. The authors show that the community assembly in their model leads, after several random invasion attempts, to species with complementarity in their top choice resource. The authors claim this finding to be “striking”, but I have some difficulties understanding why. Let us assume, for example, that two species have the same resource preference order. Since, as shown by the authors, the species spend the vast majority of their time growing on their top choice resource, one would expect that these two species were mainly competing for this resource, and so in the end the species with the highest growth rate on this top choice resource would eventually outcompete the other one (also the authors state in the manuscript that “the success of an invader depended on the growth rate on their top choice resource”). I would say, therefore, that it is not surprising that, with invasions of randomly sampled species, the ones that are left have complementary top choice resources. In fact, all other species will have gone to extinction because of competitive exclusion. Therefore, if we have in general N resources in the system one would expect that in the end N species with perfect complementarity in their top choice resource would be able to coexist. I also do not understand the choice that the authors do to compute the “null expectation” of the top choice resource complementarity. In fact, by measuring the overlap of randomly selected species they are effectively computing the overlap of an increasing number of random vectors, which will of course tend to zero as the number of vectors (i.e., species) increases. I therefore invite the authors to explain more clearly why they are claiming this findings to be striking.

We understand the reviewer’s concern about better highlighting why our results are striking. In the case of co-utilization, our theoretical results (see Supplementary Text, section I) suggest that community assembly should impose selection on the mean growth rate across all resources that a microbe can utilize, instead of just the growth rate on the top choice. In this case, the success of an invader would not depend on its top choice (i.e., the resource on which it has the highest growth rate).

Our results also show that co-utilization contrasts with diauxie, because during diauxie, selection occurs primarily on the growth rate on the top choice. We believe this is a non-

trivial finding of our work, and to the best of our knowledge, has not been previously reported.

In light of these two claims, we do believe that our findings are striking. Moreover, we do not feel that our null expectation is wrong. Prior to our results shown in Fig. 2, one could reasonably think that it was valid. Indeed, all choices other than the top choice show patterns consistent with the null expectation.

We thank the reviewer for pointing out the need to clarify the striking aspect of our results: that the top choice resource drives a disproportionate fraction of species growth, but that selection on the top choice occurs only during diauxie, not co-utilization.

In the revised manuscript, we have now added new text to clarify this aspect of our results in several places.

Lines 275-277, *“Co-utilizing microbes, instead, grow on multiple resources simultaneously, spending roughly equal time on each utilized resource.”*, 306-307, *“Such a phenomenon only occurs in diauxic species, not co-utilizing species (Supplementary Text, section I).”*

Lines 306-307, *“Such a phenomenon only occurs in diauxic species, not co-utilizing species (Supplementary Text, section I).”*

Lines 317-321, *“Selection on the top choice growth rate in diauxic communities is in striking contrast with co-utilizing communities, which we found select for the average growth rate across all resources instead (Supplementary Text, section I).”*

On the other hand, what I did find surprising, even though it is not stated explicitly in the manuscript, is the fact that contrarily to the expectation that I’ve described above two species with the same resource preference order are indeed able to coexist, e.g. if the species that grows slower on the top choice resource grows faster on the second preferred one, as shown in Fig. S8 (point I). This is, I think, a very interesting result that is “lost” in the Supplementary Information. I therefore invite the authors to highlight this finding in the manuscript.

We wholeheartedly agree with the reviewer that our model makes an important and testable prediction: that species with the same preference order can coexist. We also agree that in the original manuscript, this result was not appropriately highlighted. In the revised manuscript, we have performed a new analysis of existing experimental data to strengthen this point and highlight it in a biologically relevant context.

Specifically, we analyzed data from 13 *E. coli* strains (Barthe et. al., *mBio*, 2020), which all had the same preference order, preferring glucose to xylose. One of these strains was also anomalous, i.e., it had a higher growth rate on xylose than glucose. Using as many of the experimentally-measured parameters as were available, we performed simulations where we co-cultured all pairs of strains. We found that, indeed, several pairs of strains with the same preference order could coexist according to our simulations. This result was robust to our uncertainty in certain parameters (like lag times). Here, we randomly chose parameters in an experimentally relevant range, and found that in the majority of cases, our inference about which strains could coexist, and which would outcompete the other, remained robust. Further, in one of these cases, we observed the possibility of bistability.

We thank the reviewer for encouraging us to explore this interesting observation in the model more rigorously. We believe that it has led us to make an additional set of predictions that would be easy to test experimentally.

In the revised manuscript, we have included a new main text figure, a new section in Results, as well as new text in the “Testable Predictions” section of the Discussion, that appropriately highlights the observation in the model, as well as the predictions.

2. When talking about anomalous species (i.e., species that grow slower on their top choice resource), the authors find that in their model they are eventually always outcompeted, even if some of the assumptions that they use are relaxed (i.e., balanced resource supply). However, those anomalous species, even if not common, are consistently found in microbial communities. Is there any mechanisms that the authors have thought about that could allow such species to persist in the communities? I think it would be appropriate to add even a short comment on this matter.

This is a very pertinent question, since anomalous microbes like certain *Bacteroides* strains indeed exist in nature. We thank the reviewer for encouraging us to explore which assumptions could be relaxed to allow anomalous species to survive in the model, and not go extinct.

One way to promote the survival of anomalous microbes in our model is to add diauxic lags, as well as a “super” resource, whose mean growth rate is higher than other resources. As the diauxic lag times increase, so does the fraction of anomalous microbes that survive (~5-10% with a 1 hr lag; Fig. S8). In this scenario, a non-anomalous survivor would tend to use the super resource as its top choice. An anomalous species that prefers another resource can completely deplete it and thereby survive if the other species experiences a long enough diauxic lag before switching resources (Fig. S8, 1 hr is enough; Supplementary Text, section K). As the diauxic lag times increase, so does the survival probability of anomalous microbes.

In the revised manuscript, we have included a new supplementary figure to show results from simulations where anomalous microbes which satisfy the two criteria we described above are likely to persist in our communities. We have also added new text to highlight these examples, and connect them with other mechanisms (e.g., inhibitory interactions) that may allow some anomalous microbes to persist in natural microbial communities. We also have new analytical derivations in the supplementary text that explicitly show what conditions must be satisfied for anomalous species to coexist in such a scenario.

Changes to the text:

Lines 635-650, “*In our model, we found that anomalous species always went extinct, but anomalous microbes are indeed found in nature, albeit rarely. Examples are certain Bacteroides strains in the human gut microbiome. One way to promote the survival of anomalous microbes in our model is to add diauxic lags, as well as a “super” resource, whose mean growth rate is significantly higher (140%) than other resources. As the diauxic lag times increase, so does the fraction of anomalous microbes that survive (~5-10% with a 1 hr lag; Fig. S8). In this scenario, a non-anomalous survivor would tend to use the super resource as its top choice. An anomalous species that prefers another resource can completely deplete it and thereby survive if the other species experiences a long enough diauxic lag before switching resources (Fig. S8, 1 hr is enough).*”

Other comments:

A. The geometrical approach for the study of diauxically-growing communities, and the fact that it explains how resource depletion times are more crucial than resource concentrations for the properties and dynamics of this type of systems is a novel and very interesting contribution. It would be a really nice addition to the literature on the subject.

Thank you!

B. In the Supplementary Information, the authors show that in a system with two resources species coexistence requires the ratio R_1/R_2 of the resources' concentrations to fall within a certain range, leading e.g. to eqs (18) and (20). I think it would make things more clear and accessible if the authors added an order-of-magnitude estimate of these ranges, using some experimentally realistic values for the parameters involved.

This is an excellent suggestion. Using experimental data from 13 diauxic *E. coli* strains, we have now estimated the range of resource ratios that could enable these strains to coexist (see Supplementary Text, section J). As mentioned before, we have also performed new simulations with these realistic parameter values in our model, and found a range of possible outcomes that are possible in pairwise co-cultures, such as coexistence, competitive exclusion, and bistability.

C. In the Supplementary Information, the authors state that “In presence of an anomalous bug, a bistability may arise”. I think this is an important prediction of their framework, which is however absent from the manuscript. I therefore invite the authors to add this prediction to the ones they are already mentioning in the manuscript.

We have added our prediction regarding bistability to the “Testable Predictions” section of the revised manuscript. We have also performed new analyses showing bistability arising even without anomalous species, and can be induced by lag times, for example.

D. There are a few typos throughout the manuscript and the Supplementary Information. I invite the authors to check and correct them.

Thank you for pointing this out. We have corrected several typos in the manuscript (e.g., repeating words in captions of Fig. 5 in the original manuscript) and have run the revised manuscript and supplementary information through spell checking software.

Reviewers' Comments:

Reviewer #1:

Remarks to the Author:

The authors carefully addressed all the points made previously. The new version of the manuscript better highlights the biological relevance of the work. The new analyses with *E. coli* and *Bacteroides* strains are added value. I think the paper has improved significantly and recommend it for publication.

Reviewer #2:

Remarks to the Author:

I am very satisfied with the author's responses.

They have addressed all my concerns satisfactorily, and I think the manuscript has now improved.

I recommend the publication of the manuscript in its current form.

REVIEWERS' COMMENTS

Reviewer #1 (Remarks to the Author):

The authors carefully addressed all the points made previously. The new version of the manuscript better highlights the biological relevance of the work. The new analyses with *E. coli* and *Bacteroides* strains are added value. I think the paper has improved significantly and recommend it for publication.

We are grateful to the reviewer for their encouragement and positive evaluation of our work, and thank them for their original suggestions which helped improve the manuscript.

Reviewer #2 (Remarks to the Author):

I am very satisfied with the author's responses.

They have addressed all my concerns satisfactorily, and I think the manuscript has now improved.

I recommend the publication of the manuscript in its current form.

We thank the reviewer for their feedback and encouragement, and for their confidence in our work. Their original comments helped strengthen our paper.